# The photosystem I assembly apparatus consisting of Ycf3–Y3IP1 and Ycf4 modules

Sreedhar Nellaepalli[1,2], Shin-Ichiro Ozawa[1,2], Hiroshi Kuroda[1,2] & Yuichiro Takahashi[1,2]

In oxygenic photosynthesis, light energy is converted into redox energy by two photosystems (PSI and PSII). PSI forms one of the largest multiprotein complexes in thylakoid membranes consisting of a core complex, peripheral light-harvesting complexes (LHCIs) and cofactors. Although the high-resolution structure of the PSI–LHCI complex has been determined, the assembly process remains unclear due to the rapid nature of the assembly process. Here we show that two conserved chloroplast-encoded auxiliary factors, Ycf3 and Ycf4, form modules that mediate PSI assembly. The first module consists of the tetratricopeptide repeat protein Ycf3 and its interacting partner, Y3IP1, and mainly facilitates the assembly of reaction center subunits. The second module consists of oligomeric Ycf4 and facilitates the integration of peripheral PSI subunits and LHCIs into the PSI reaction center subcomplex. We reveal that these two modules are major mediators of the PSI–LHCI assembly process.

[1] Research Institute for Interdisciplinary Science, Okayama University, 3-1-1 Tsushima-naka, Kita-ku, Okayama 700-8530, Japan. [2] JST-CREST, Tokyo, Japan. Correspondence and requests for materials should be addressed to Y.T. (email: taka@cc.okayama-u.ac.jp)

In oxygenic photosynthesis, photosystem I (PSI) mediates the light-induced electron transfer from plastocyanin or cytochrome $c$ to ferredoxin. The PSI core complex exists as a homotrimer in cyanobacteria and as a monomer in plants and algae, but the main constituent subunits are well conserved among these photosynthetic organisms[1]. In vascular plants, PSI harbors 15 core and 4 LHCI subunits, and 155–156 Chls (143 Chls $a$, 12–13 Chls $b$), 35–36 carotenoids, 10–32 lipids, three 4Fe-4S clusters, and two phylloquinones[2–4]. A green alga, *Chlamydomonas reinhardtii*, contains fourteen PSI core and nine LHCI subunits[5,6]. PsaA-C and PsaJ are chloroplast-encoded while the other core subunits (PSAD-PSAI, PSAK-L, PSAN, and PSAO) and LHCIs (LHCA1-9) are nuclear-encoded[7]. Thus, the biogenesis of PSI complex in plants and algae involves a coordinated synthesis of nuclear- and chloroplast-encoded polypeptides along with the cofactors, thereby mediated by a sequential multistep assembly process. However, it has been difficult to analyze the assembly process since the PSI complex biogenesis proceeds rapidly[8]. Several auxiliary factors such as hypothetical chloroplast open reading frame 3 (Ycf3)[9], Ycf4[9], Pale Yellow Green 7 (PYG7)[10], Ycf3-interacting protein 1 (Y3IP1)[11], PSBP-Domain Protein 1 (PPD1)[12], Photosystem I Assembly 2 (PSA2)[13] and Photosystem I Assembly 3 (PSA3)[14], which post-translationally assist PSI biogenesis, have been reported so far[15,16]. Among them, two chloroplast-encoded proteins, Ycf3 and Ycf4, as well as nuclear-encoded PYG7 (Conserved in Green Lineage 71 (CGL71)[17] and Ycf37[18] are orthologues in *C. reinhardtii* and cyanobacteria, respectively), are conserved in photosynthetic organisms[9,16,19] and play important roles in PSI complex assembly.

Ycf3 is an extrinsic protein associated with the thylakoid membranes and is essential for PSI complex biogenesis in *C. reinhardtii*[9] and tobacco[20]. This contains three tetratricopeptide repeat (TPR) domains, which could be involved in mediating the interaction between proteins. Ycf3 has an affinity with solubilized PSI complexes and more specifically has an interaction with PsaA and PSAD in vitro[21]. It is thus proposed that Ycf3 may be involved in an initial assembly of PSI proteins[9,21]. Y3IP1 has a putative transmembrane helix and copurified with Ycf3 in tobacco[11]. In tobacco, knockdown of *Y3IP1* resulted in a specific deficiency in PSI but did not impair the accumulation of Ycf3[11]. CGL59 is orthologous of Y3IP1 in *C. reinhardtii*[22]. Ycf4 contains two putative transmembrane helices at its N-terminal region and is present in the thylakoid membranes. Knockout of *ycf4* resulted in no accumulation of PSI complex in *C. reinhardtii*[9], while Ycf4 is not essential but important in cyanobacteria[19] and tobacco[23] because a smaller amount of PSI accumulates in the knockout mutants. Affinity purification of tagged Ycf4 showed its interactions with newly synthesized PSI proteins in *C. reinhardtii*[24]. However, our knowledge of how these factors are involved in the PSI assembly at the molecular level is limited.

Here we describe the evidence that three auxiliary factors, Ycf3, Y3IP1, and Ycf4, form a core PSI assembly apparatus, with which PSI complex assembly intermediates were copurified. For this purpose, we expressed Ycf3 or Ycf4 fused with HA (human influenza hemagglutinin) tag in the chloroplast by the chloroplast expression system[25], or expressed Y3IP1 fused with HA tag by the nuclear expression system[26,27] in *C. reinhardtii*. Subsequently, we carried out affinity purification of these HA-tagged proteins from the solubilized thylakoid membranes and characterized interacting proteins including PSI assembly intermediates.

## Results

### Generation of chloroplast mutants expressing HA-tagged Ycf3.
To generate mutants expressing Ycf3 fused with HA-tag at the C-terminus (Ycf3-HA), we constructed the chloroplast transforming vector in which the *ycf3-HA* expression cassette[25,28] had been inserted at the *Bam*HI site in downstream of the *psbA* gene (Fig. 1a). The resulting vector was delivered by a particle gun into a chloroplast mutant, Fud7, which has a deletion in the *psbA* gene[29]. Putative transformants were selected under photoautotrophic conditions, and their genotype was confirmed by the amplification of the corresponding chloroplast DNA region by PCR (Supplementary Fig. 1a). We also obtained ycf3-HA/Δycf3 double mutants by transforming the ycf3-HA mutant with the chloroplast vector in which the *ycf3* gene has been disrupted by the *aadA* cassette as already described[9].

The ycf3-HA mutants accumulated both Ycf3 and Ycf3-HA at a comparable level, whereas the ycf3-HA/Δycf3 double mutant contained only Ycf3-HA as expected (Fig. 1b). These two mutants accumulated Ycf4 and a diagnostic PSI core protein (PSAF) at WT levels. The ycf3-HA mutants, as well as ycf3-HA/Δycf3 mutant, showed PSI activity according to the fluorescence induction kinetics measurements (Supplementary Fig. 1b) and photoautotrophic growth, albeit at slightly slow rates (Supplementary Fig. 1c). These observations indicate that the Ycf3-HA is sufficiently active in the assembly of functional PSI complex.

### Copurification of Y3IP1 and PSI RC subunits with Ycf3-HA.
To identify Ycf3-HA interacting proteins, we isolated the thylakoid membranes from the ycf3-HA mutant, solubilized with α-DM and the thylakoid extracts were applied to affinity spin column. The resulting Ycf3-HA preparation contained two distinct stained polypeptides corresponding to Ycf3-HA and Y3IP1 in similar amounts, although the preparation from the WT thylakoid extracts as a negative control exhibited only minor non-specific proteins around 50–65 kDa (Fig. 1c). Immunoblotting confirmed the presence of Ycf3-HA and Y3IP1 as expected and also detected endogenous Ycf3 and Ycf4, indicating for the first time the interaction among Ycf3, Y3IP1, and Ycf4 (Fig. 1d). Since the endogenous Ycf3 and the Ycf4 were scarcely stained on the gel, their amounts were much lower than those of Ycf3-HA and Y3IP1. PSI proteins, PsaA and PSAF, were also identified, whereas no signal of PsaC and PSAD was detected (Fig. 1d). LC-MS/MS detected a peptide fragment from the other reaction center (RC) subunit, PsaB, in the Ycf3-HA preparation, suggesting that both PsaA and PsaB were separated around 66 kDa on the gel (Supplementary Table 1).

### Ycf3–Y3IP1 assists assembly of PSI RC subunits.
Next, we purified Ycf3-HA from the cells of which total cellular proteins had been pulse-labeled with $^{35}$S to address whether the PSI proteins copurified with Ycf3-HA are newly synthesized. Autoradiography of the thylakoid proteins revealed that D1 (a PSII RC protein), LHCII proteins (Type-I, III, and IV), and ATP synthase proteins (AtpA and AtpB) were mainly pulse labeled and the labels were stable after 6 h chase (Fig. 1e). In contrast, pulse-labeled PsaA/PsaB (around 66 kDa) were markedly enriched in the Ycf3-HA preparation and were significantly decreased to ~30% of the pulse-labeled level after 6 h chase, indicating that the Ycf3-HA transiently bound newly synthesized PsaA/PsaB (Fig. 1e). In contrast, the labeling of PSAF was not observed. Ycf3, which has four sulfur-containing amino acid residues including the N-terminal Met, was pulse-labeled approximately threefold more than Y3IP1, of which mature form has ten sulfur-containing amino acid residues. Thus, the synthesis of Ycf3 was estimated to be 7–8 times faster than that of Y3IP1. However, the labeled Ycf3-HA decreased to ~20% of the pulse-labeled level, as fast as the labeled PsaA/PsaB, whereas the labeled Y3IP1 was rather stable (Fig. 1e). Thus, it is inferred that the rapid turnover

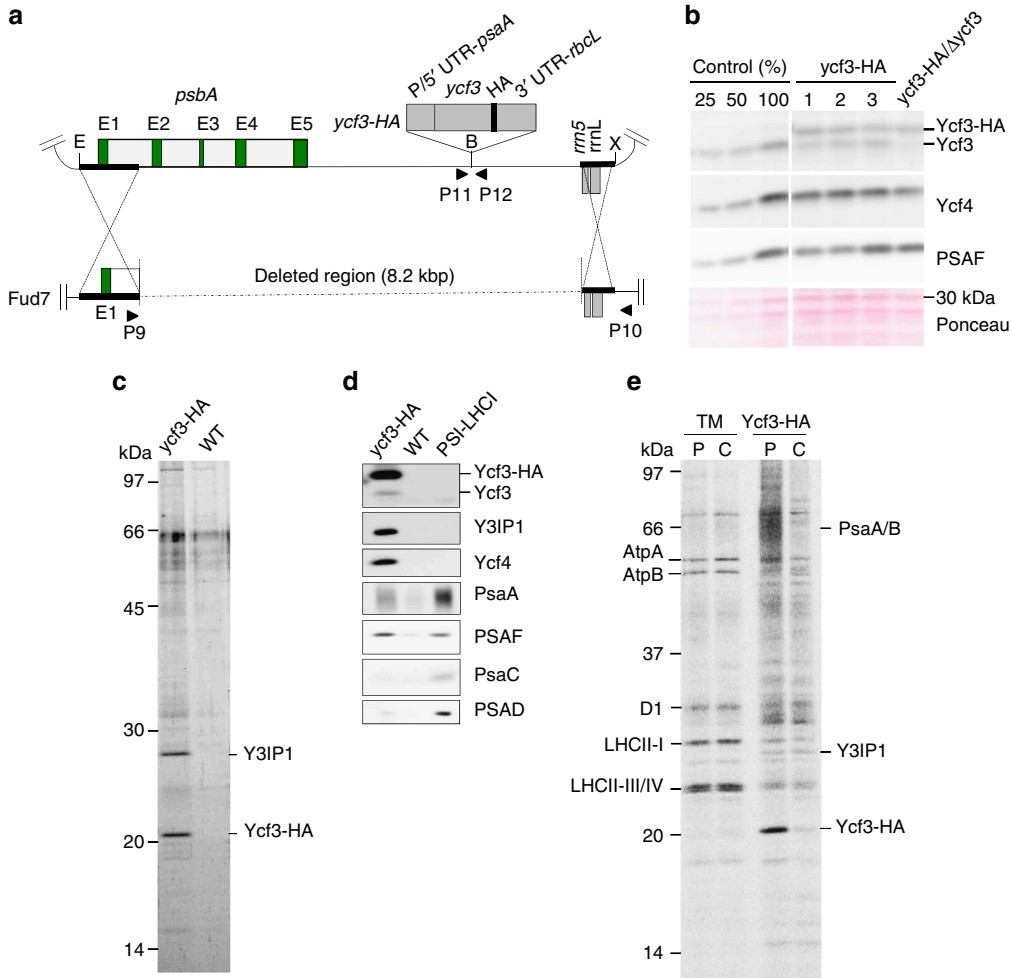

**Fig. 1** Ycf3–Y3IP1 transiently associates with newly synthesized PSI reaction center subunits. **a** Physical map of the vector containing *ycf3-HA* expression cassette inserted at the *Bam*HI site downstream of the *psbA* gene, which consists of five exons (E1–E5), is shown. The host strain, Fud7, has an 8.2 kbp deletion starting from the downstream of the E1. The *ycf3* cassette contains the coding sequence of *ycf3* and *HA* tag flanked by P/5′-UTR of *psaA* and 3′-UTR of *rbcL*. Ycf3-HA contains HA tag inserted at C-terminus of Ycf3. E, X, and B depict restriction sites for *Eco*RI, *Xho*I, and *Bam*HI, respectively. P9/P10 and P11/P12 represent pairs of primers used for PCR (Supplementary Fig. 1a). **b** The expression of Ycf3-HA was confirmed by immunoblotting. Total cellular proteins from control, ycf3-HA (clones 1–3), and ycf3-HA/Δycf3 mutants grown under photoheterotrophic condition were analyzed. The nitrocellulose membrane was stained with Ponceau for loading control. **c** Polypeptide compositions of the affinity-purified preparations from ycf3-HA and WT strains. Polypeptides separated by SDS-PAGE were visualized by staining with the fluorescence dye, Flamingo. Ycf3-HA and Y3IP1 were detected in the Ycf3-HA preparation. **d** Immunoblotting with antibodies against Ycf3, Y3IP1, Ycf4, and PSI polypeptides (PsaA, PsaC, PSAD, and PSAF). The purified PSI–LHCI was loaded as a reference. **e** Pulse-chase labeling of proteins with $^{35}$S. Cells were labeled for 10 min (P) and then were chased for 6 h (C). TM: the purified thylakoid membranes, Ycf3-HA: the affinity-purified Ycf3-HA

of Ycf3 maintained the stoichiometric accumulation of Ycf3 and Y3IP1 (Fig. 1c).

To address whether PSI RC subunits are assembled into a PSI RC subcomplex, the concentrated Ycf3-HA preparation was fractionated by sucrose density gradient (SDG) ultracentrifugation (Fig. 2). Immunoblotting revealed that most Ycf3-HA and Y3IP1 were detected in the upper region of SDG, indicating that the interaction between Ycf3–Y3IP1 complex and PSI subunits was unstable. PSI subunits were mainly separated in fractions around 700 kDa region, where most Ycf4 and a small amount of Ycf3-HA were detected. It is inferred from the estimated size of the PSI protein on SDG that a PSI RC subcomplex (PsaA/PsaB heterodimer), to which Ycf4 and a small amount of Ycf3-HA may bind, was formed (Fig. 2). The PSAF in Ycf3-HA preparation was detected as weak signals in the fractions near the bottom of SDG, suggesting that the PSAF was not yet integrated into the major PSI RC subcomplex. It was difficult to characterize the fractions containing PSAF as the amounts of the proteins were quite low.

To provide insights into the function of CGL59/Y3IP1, we obtained a knockout mutant of *Y3IP1* (ΔY3IP1), which has an insertion of the paromomycin resistance cassette (CIB1) at the 7th exon of *CGL59* (Fig. 3a), from Chlamydomonas Library Project (CLiP)[30]. This mutant was unable to grow photoautotrophically and accumulated PSI and Ycf3 at ~5% and ~30% of CLiP host strain (control) level, respectively (Fig. 3b, c). Consequently, ΔY3IP1 lacked a detectable level of PSI activity (Supplementary Fig. 2). It is inferred that Y3IP1 plays an important role in the accumulation of Ycf3 and PSI.

Complementation of the ΔY3IP1 mutant with cDNA of *Y3IP1* or *Y3IP1-HA* by the nuclear expression system[26,27] restored photoautotrophic growth (Fig. 3b), accumulation of PSI proteins (Fig. 3c), and PSI activity (Supplementary Fig. 2) like the control strain. The thylakoid membranes were isolated and solubilized with α-DM, and the thylakoid extracts were applied onto spin column for the affinity purification. Figure 3d compares the polypeptides of the Y3IP1-HA and Ycf3-HA preparations. The

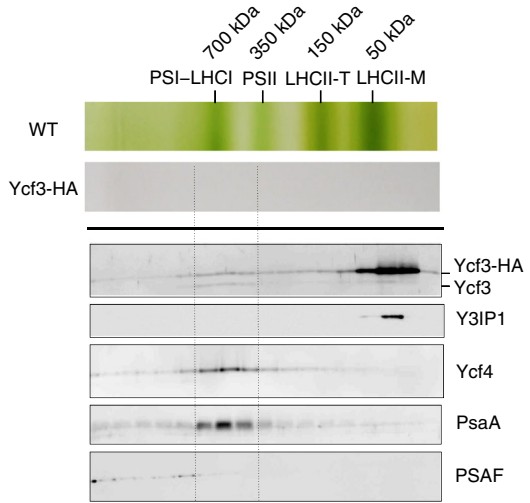

**Fig. 2** Separation of Ycf3-HA preparation on SDG ultracentrifugation. The purified Ycf3-HA was concentrated and subjected to SDG ultracentrifugation, and the resulting gradient was fractionated and was analyzed by immunoblot using anti-PsaA, Y3IP1, Ycf4, Ycf3, and PSAF antibodies. Ycf3 and Y3IP1 were fractionated in the upper region of the gradient, whereas PsaA and Ycf4 were detected at positions around 700 kDa. Separation of chlorophyll–protein complexes from the WT thylakoid extracts was shown on the top as a reference to show apparent molecular sizes on the gradient. LHCII-T: trimeric LHCII, LHCI-M: monomeric LHCII

Y3IP1-HA preparation contained both Y3IP1-HA and Ycf3 although Ycf3 is reduced with respect to Y3IP1. Immunoblotting revealed that PsaA and Ycf4 were also present in the Y3IP1-HA preparation (Fig. 3e). These observations confirmed that the Y3IP1 and Ycf3 have a stable interaction and form a Ycf3–Y3IP1 complex.

**Generation of chloroplast mutants expressing HA-tagged Ycf4.** In the previous report, we employed affinity purification of a Ycf4 complex by TAP (tandem affinity purification)-tag technology and detected some interacting PSI subunits (PsaA, PsaB, PsaC, PSAD, PSAE, and PSAF) in the preparation[24]. Since PSAD and PSAF copurified with Ycf4-TAP were newly synthesized, we concluded that the Ycf4 complex acts as the scaffold for PSI assembly. However, the decreased accumulation of Ycf4 to 25% of WT level by fusion of TAP-tag at its C-terminus and the low yield of TAP-tagged Ycf4 preparation by two-step affinity purification hampered detailed investigations on the molecular mechanism of how Ycf4 assists PSI core complex assembly.

In the present study, we generated HA-ycf4 mutants using the same strategy for generating ycf3-HA mutants (Fig. 4a). Putative transformants were selected under photoautotrophic conditions, and their genotype was confirmed by the amplification of the corresponding chloroplast DNA region by PCR reaction (Supplementary Fig. 3a). We also obtained HA-ycf4/Δycf4 double mutants by transforming HA-ycf4 mutant with the chloroplast vector in which the ycf4 gene has been disrupted by the aadA cassette[9]. It was shown that HA-Ycf4 accumulated about 20-fold

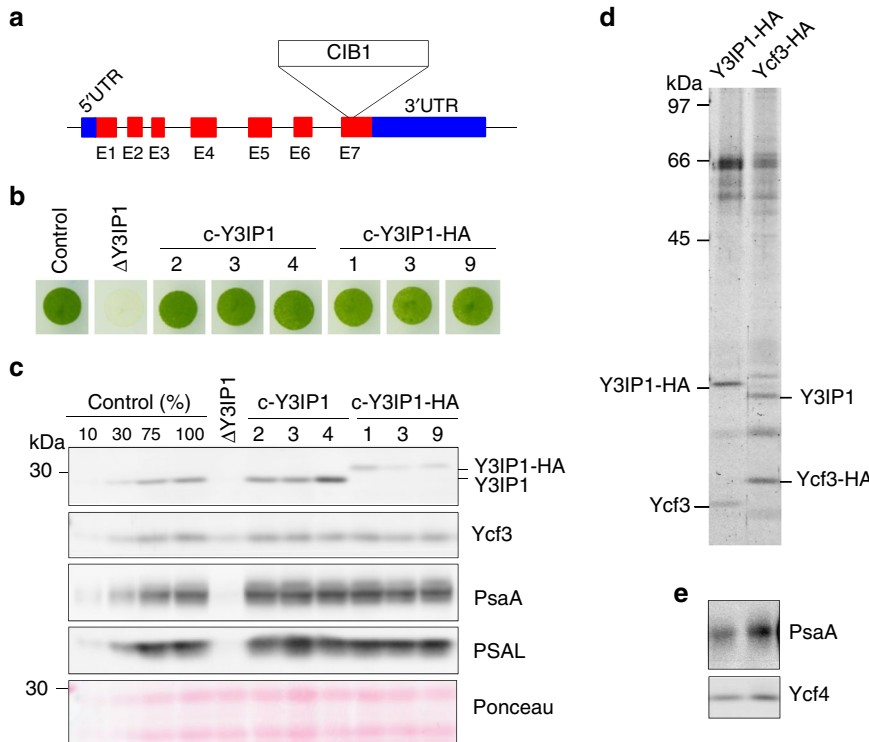

**Fig. 3** Affinity purification of Y3IP1-HA. **a** ΔY3IP1/CGL59 mutant (LMJ.RY0402.195677) from the CLiP contains the paromomycin resistance cassette (CIB1) inserted at the 7th Exon (E7) of CGL59/Y3IP1 (Cre06.g280650). Exons, introns, and untranslated regions are shown as red boxes, black lines, and blue boxes, respectively. This strain also contains second CIB1 cassette in CAH2 locus (Cre04.g223050). The mutant was complemented with cDNA of Y3IP1 (c-Y3IP1) or Y3IP1-HA (c-Y3IP1-HA). **b** The growth of CLiP host strain as control, ΔY3IP1, and three c-Y3IP1 clones (2, 3, and 4) and c-Y3IP1-HA clones (1,3, and 9), under the photoautotrophic condition in the light of 50 μmol photons m$^{-2}$ s$^{-1}$. **c** Total cell proteins from control, ΔY3IP1, c-Y3IP1, and c-Y3IP1-HA strains were analyzed by immunoblotting using antibodies against Y3IP1, Ycf3, PsaA, and PSAL. The nitrocellulose membrane was stained with Ponceau for loading control. **d** The polypeptide composition of the affinity-purified Y3IP1-HA and Ycf3-HA. Polypeptides separated by SDS-PAGE were visualized by staining with Flamingo. **e** Immunoblotting of the purified Y3IP1-HA and Ycf3-HA with antibodies against PsaA and Ycf4

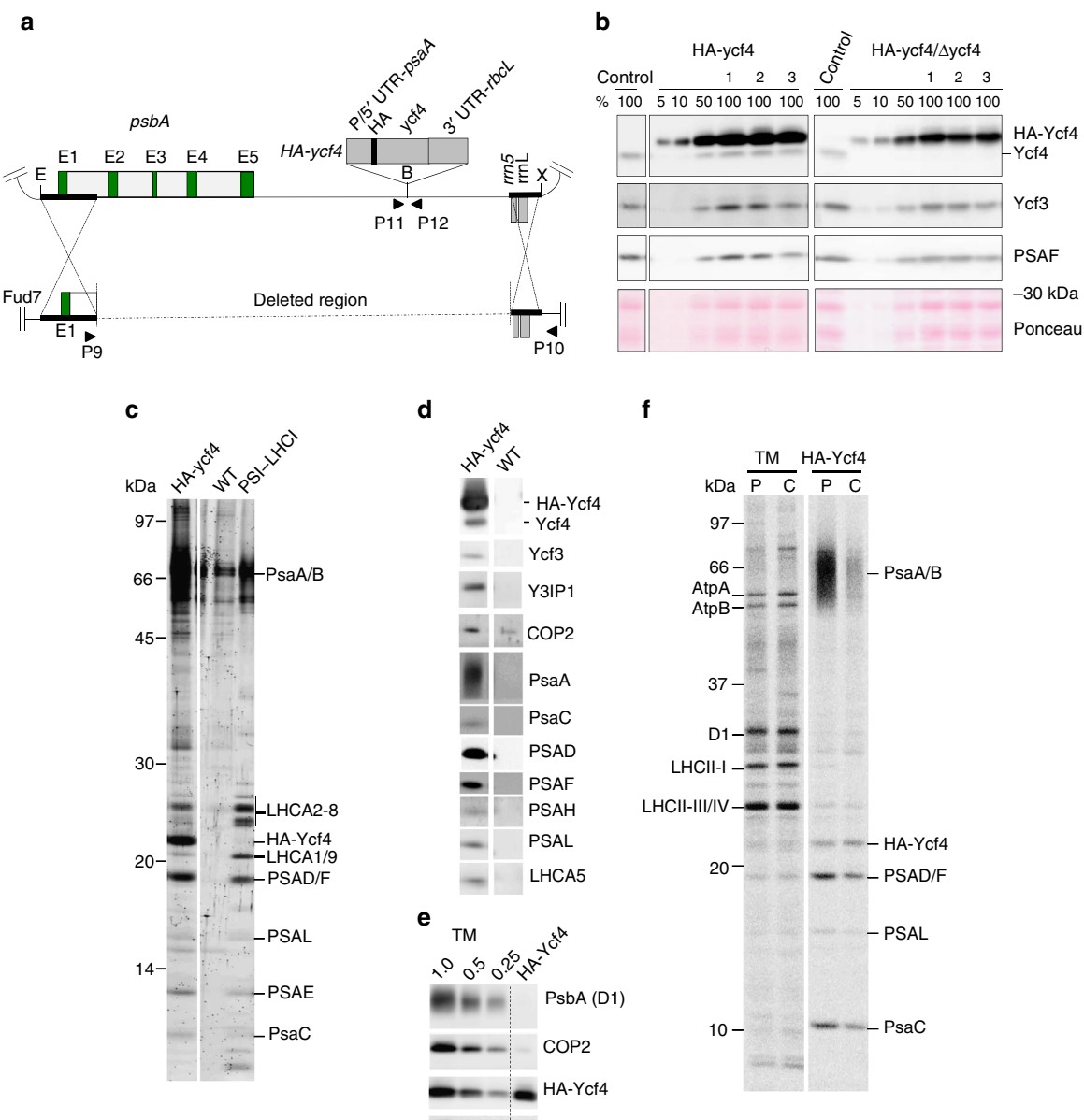

**Fig. 4** HA-Ycf4 transiently associates with newly synthesized PSI core subunits. **a** Physical map of the vector containing *HA-ycf4* expression cassette inserted at the *Bam*HI site downstream of the *psbA* gene is shown. The *ycf4* cassette contains the coding sequence of *HA* tag and *ycf4* flanked by P/5'-UTR of *psaA* and 3'-UTR of *rbcL*. HA-Ycf4 contains HA tag inserted at N-terminus of Ycf4. E, X, and B depict restriction sites for *Eco*RI, *Xho*I, and *Bam*HI, respectively. P9/P10 and P11/P12 represent pairs of primers used for PCR (Supplementary Fig. 3a). **b** The expression of HA-Ycf4 was confirmed by immunoblotting. Total cellular proteins from control, HA-ycf4 (clones 1–3), and HA-ycf4/Δycf4 (clones 1–3) mutants grown under photoheterotrophic condition were analyzed. The nitrocellulose membrane was stained with Ponceau for loading control. **c** Polypeptide compositions of the affinity-purified preparations from HA-ycf4 and WT strains. The purified PSI–LHCI was loaded as a reference. Polypeptides separated by SDS-PAGE were visualized by staining with Flamingo. **d** Polypeptide compositions detected by immunoblotting. **e** Estimation of enrichment of COP2 in the HA-Ycf4 preparation. Dilution series of thylakoid membranes (TM) corresponding to 1.0, 0.5, and 0.25 μg Chls per lane were loaded to estimate the abundance of D1, COP2, HA-Ycf4, and PSAD in the HA-Ycf4 preparation by immunoblotting. **f** Pulse-chase labeling of proteins with $^{35}$S. Cells were pulse-labeled for 10 min (P) and then were chased for 6 h (C). TM; the purified thylakoid membranes, HA-Ycf4; the affinity-purified HA-Ycf4 preparation

more in HA-ycf4 and HA-ycf4/Δycf4 double mutants than that of the endogenous Ycf4 in the control strain (Fig. 4b). The HA-ycf4 mutant accumulated Ycf3 and PSAF as well as the endogenous Ycf4 at normal levels (Fig. 4b) and showed PSI activity and photoautotrophic growth (Supplementary Fig. 3b, c), indicating that the over-expression of HA-Ycf4 had no deteriorating effect on functional assembly of PSI complex. The HA-ycf4/Δycf4 mutant also accumulated PSI and grew photosynthetically similar to control strain (Fig. 4b, Supplementary Fig. 3b, c), indicating

that the HA-Ycf4 is functional. Ycf4 is part of a large complex and, when solubilized, is separated into bottom fractions of SDG[9]. To confirm whether HA-Ycf4 retains its ability to form a large complex, extracts of the thylakoid membrane from HA-ycf4 and HA-ycf4/Δycf4 mutants were separated by SDG ultracentrifugation (Supplementary Fig. 4). As expected, PsaA was mainly detected in A3 fraction (PSI–LHCI supercomplex, 700 kDa) and was slightly present in the heavier fractions (PSI–LHCI/II supercomplex)[31]. It was shown that HA-Ycf4 was mainly

separated in heavier fractions as the endogenous Ycf4, confirming that the overexpressed HA-Ycf4 forms a large complex. In the following experiments, we used the HA-ycf4 mutant for affinity purification of HA-Ycf4 complexes.

**Copurification of PSI core and LHCI proteins with HA-Ycf4.** We purified HA-Ycf4 from the HA-ycf4 mutant by the method as described for Ycf3-HA purification except that the thylakoid membranes were solubilized with β-DM. The purification completed within 2 h although it took over 24 h to purify Ycf4-TAP in the previous study[24]. The resulting preparation contained several proteins of around 25 and 66 kDa (Fig. 4c). Immunoblotting detected not only HA-Ycf4 but also endogenous Ycf4 (Fig. 4d), which indicates an interaction of Ycf4 with HA-Ycf4 and supports the existence of an oligomeric form of Ycf4. Small amounts of Ycf3 and Y3IP1 were also detected in the HA-Ycf4 preparation (Fig. 4d). Immunoblotting and LC-MS/MS analyses assigned 66 kDa bands as PsaA and PsaB (Fig. 4d and Supplementary Table 1) and revealed the presence of six peripheral PSI core subunits (PsaC, PSAD-F, PSAH, PSAL) as well as some LHCI subunits (Fig. 4d). The two-step affinity-purified TAP-tagged Ycf4 (Ycf4-TAP) preparation contained PSAF and the Chlamyopsin protein (COP2) as major proteins, and retained smaller amounts of PsaA, PsaB, PsaC, PSAD, and PSAE, but no LHCI proteins[24]. A weak signal of COP2 was detected not only in the HA-Ycf4 preparation but also in the preparation from WT (Fig. 4d). To evaluate whether COP2 is a specifically interacting protein in the purified HA-Ycf4 preparation, a dilution series of thylakoid membranes were compared with affinity-purified HA-Ycf4 by probing with antibodies against D1, COP2, HA, and PSAD (Fig. 4e). It was revealed that D1 and COP2 were not enriched, whereas HA-Ycf4 and PSAD were enriched in the HA-Ycf4 preparation. Since the purification condition in the present study was much milder than the previous study, it is unlikely that COP2 was lost during the purification. When the thylakoid extracts were incubated with the affinity beads overnight, more non-specific proteins including COP2 were copurified. Thus, it is likely that the copurification of COP2 with the TAP-tagged Ycf4 was caused by the long incubation of the thylakoid extracts with the affinity beads.

**Ycf4 assists integration of PSI core and LHCI proteins.** When HA-Ycf4 was purified from cells of which total cellular proteins had been pulse labeled, the chloroplast-encoded PsaA, PsaB, and PsaC, as well as the nuclear-encoded PSAD, PSAF, and PSAL were labeled. PSAE was not labeled because of the lack of Met and Cys (Fig. 4f). Although the labels of PsaA and PsaB were barely observed in the thylakoid membranes, these signals were remarkably enriched in the HA-Ycf4 preparation, indicating that the PSI proteins copurified with HA-Ycf4 were newly synthesized. The labels of these PSI proteins decayed after 6 h chase, indicating that newly synthesized PSI proteins are transiently bound to the Ycf4-HA preparation. Interestingly, the turnover of the PSI RC subunits was faster than that of the peripheral PSI subunits (PsaC, PSAD, and PSAF). The pulse-labeled RC subunits were decreased to ~30% after 6 h chase, whereas the pulse-labeled peripheral subunits were reduced to ~50%. Since PSI core subcomplex was assembled rapidly on the Ycf4 preparation, it is expected that the labels of the RC and peripheral subunits would decrease at similar rates. The difference in the stability of the labels suggests that the synthesis of PSI RC subunits and their integration into the PSI RC subcomplex are strictly coupled, whereas the peripheral PSI subunits may remain unintegrated for a while after the synthesis in the chloroplast or cytosol. As a result, a mixture of unlabeled and labeled peripheral PSI subunits transiently accumulates and a

small amount of labeled peripheral PSI subunits may be integrated into the PSI RC subcomplex during the chase period. This results in the slower decay in the labels. The temporal difference in the integration of the newly synthesized peripheral subunits reflects the stepwise assembly process; the initial assembly of the RC subunits and the subsequent integration of the peripheral subunits.

In contrast to PSI core proteins, LHCI proteins, which contain several Met and Cys residues, were scarcely labeled (Fig. 4f). It is thus inferred that LHCI complexes, which pre-exist stably in the thylakoid membranes, are assembled into a newly synthesized PSI core complex on the HA-Ycf4 complex. In TAP-tagged Ycf4 affinity preparation, LHCI subunits were not detected[24], which may be due to destabilization of LHCI complexes by the large TAP tag fused with Ycf4 (28 kDa) or the longer and harsher purification procedure.

Pulse-labeled HA-Ycf4 was detected and was stable for 6 h chase. Interestingly, it was estimated that Ycf4 accumulates more than Ycf3 although the chloroplast ycf4 and ycf3 genes are co-transcribed and are expected to be translated at similar rates[9]. This contradiction can be explained by the present results that Ycf4 is stable while Ycf3 turns over as shown in Fig. 1e.

The purified HA-Ycf4 complex was green, and its absorption spectrum suggested the presence of chlorophylls and carotenoids (Supplementary Fig. 5a). The HA-Ycf4 complex showed the absorption peak at 676 nm, which is similar to that of the purified PSI–LHCI preparation (the absorption peak at 678 nm). HPLC analysis showed the presence of Chl *a*, Chl *b*, and β-carotene in the HA-Ycf4 complex (Supplementary Fig. 5b). These data strongly suggest that the PSI proteins present in the HA-Ycf4 preparation are associated with pigments to form a holocomplex.

To confirm whether PSI and LHCI subunits in the HA-Ycf4 complex assemble into a PSI subcomplex, the affinity-purified HA-Ycf4 complex was concentrated and then fractionated by SDG ultracentrifugation. Figure 5a shows that the chlorophylls in the HA-Ycf4 preparation were separated into two green bands, which are tentatively designated as Y1 and Y2, at the positions of A2 (PSI core complex) and A3 (PSI–LHCI supercomplex) from the WT thylakoid extracts, respectively. In contrast, HA-Ycf4 was separated broadly in Y1, Y2, and Y3 (Supplementary Fig. 6), indicating that the purified HA-Ycf4 remains its large oligomeric structure although the structure appears to be more heterogeneous in size. It is likely that the separated PSI core and PSI–LHCI subcomplexes may be dissociated from the oligomeric HA-Ycf4 according to the apparent sizes on SDG. Figure 5b–d shows that Y1 contained most PSI core proteins; PsaA-C, PSAD-F, PSAL and PSAI but lacked PSAG and PSAK, suggesting that Y1 is a PSI core subcomplex similar to the assembly intermediate detected in WT cells at logarithmic growth phase[8]. In contrast, Y2 contained both the PSI core and LHCI proteins except for LHCA2 and LHCA9. Thus, the PSI in Y2 corresponds to a new assembly intermediate, PSI–LHCI subcomplex, in which PSAG and PSAK, as well as LHCA2 and LHCA9, remain to be integrated. It appears that PSAF present in the HA-Ycf4 complex was integrated to PSI core and PSI–LHCI subcomplexes although PSAF in Ycf3-HA preparation was not yet integrated to the PSI RC subcomplex as shown in Fig. 2.

## Discussion

PSI complex is one of the largest multiprotein complexes in the thylakoid membranes, but the molecular mechanism for PSI complex assembly remain elusive. It has been difficult to characterize the assembly mechanism because PSI complex assembly proceeds rapidly and assembly intermediates accumulate in small amounts[8]. In the present study, we have successfully used

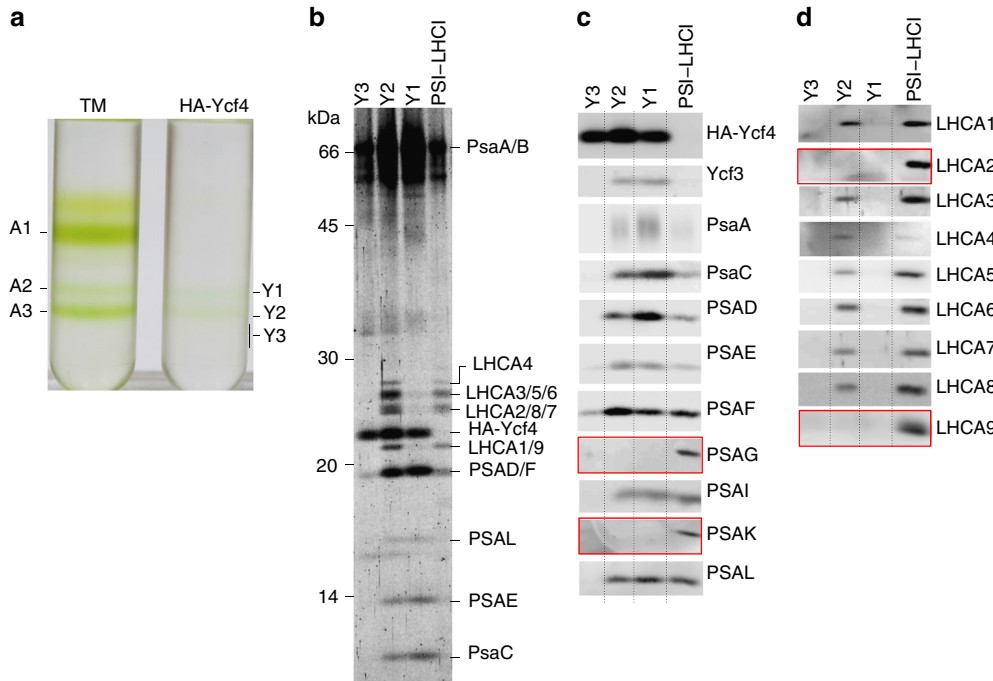

**Fig. 5** Ycf4 oligomer assists the assembly of PSI core and PSI–LHCI subcomplexes. **a** SDG ultracentrifugation separated three chlorophyll–protein complexes, A1, A2, and A3, from WT thylakoid membrane extracts (TM) while two PSI assembly intermediates, Y1 and Y2, from the HA-Ycf4 preparation (see also Supplementary Fig. 6). Y1 and Y2 are green, whereas Y3 is colorless. The positions of Y1 and Y2 correspond to those of A2 (PSII core) and A3 (PSI–LHCI), respectively. **b** Polypeptides of Y1, Y2, and Y3, after concentration, were separated by SDS-PAGE and stained with Flamingo. **c** Immunoblotting of Y1, Y2, and Y3 with antibodies against Ycf3, Ycf4, and various PSI proteins. **d** Immunoblotting of Y1, Y2, and Y3 with antibodies against nine LHCI proteins. Red frames indicate the polypeptides that are present in PSI–LHCI but absent in Y1 and Y2

chloroplast and nuclear mutants in which either HA-tagged Ycf3, Ycf4, or Y3IP1 is expressed in the chloroplast of *C. reinhardtii*, to elucidate the functions of the assembly factors and to identify PSI assembly intermediates which are expected to interact with these factors.

Affinity purifications of Ycf3-HA and Y3IP1-HA independently revealed that Ycf3 and Y3IP1 associate with each other to form a Ycf3–Y3IP1 module (Figs. 1 and 2). This module bound a substoichiometric amount of PSI RC subunits, PsaA and PsaB. The interaction between Ycf3 and PsaA has already been reported using denatured thylakoid membrane proteins bound to a nitrocellulose filter[21]. However, the present study revealed that the Ycf3–Y3IP1 module transiently binds newly synthesized RC subunits, indicating that this module plays an essential role in an initial step of the PSI RC assembly. It was reported that the solubilized Ycf3 also has an interaction with the extracted PSAD in vitro[21]. However, PSAD was not detected in the Ycf3 preparation, suggesting that the affinity between the Ycf3–Y3IP1 module and PSAD, if any, must be very weak. Co-fractionation of PSI RC along with Ycf4 on SDG at 700 kDa revealed a stable interaction between them (Fig. 2), suggesting that Ycf4 readily bound to the PSI RC subcomplex upon the assembly of PSI RC subunits assisted by the Ycf3–Y3IP1 module. PSAF in the Ycf3–Y3IP1 module was not pulse-labeled (Fig. 1c–e) nor associated with the major PSI RC fraction (Fig. 2). It was reported that PSAF, which accumulated at a low level in ΔPsaA/B mutant, was associated with Ycf4[24], suggesting that the PSAF in the Ycf3-HA preparation is preferentially bound to the copurified Ycf4 and is not yet assembled into the PSI RC subcomplex. Since the amount of PSI RC subunits present in the Ycf3–Y3IP1 module was very low, it was difficult to show the presence of cofactors in the preparation. Thus, it is important to make a large-scale preparation to allow for detailed biochemical characterization as a future study.

Of interest, the oligomeric HA-Ycf4 module associated more PSI proteins than the Ycf3–Y3IP1 module. The oligomeric Ycf4 module, to which newly synthesized PSI proteins were transiently bound, is involved in PSI core complex assembly steps after the initial PSI RC assembly assisted by the Ycf3–Y3IP1 module. Thus, we proposed that PSI core subcomplex was assembled by integrating peripheral PSI proteins (except for PSAG and PSAK) into the PSI RC subcomplex by the assist of the oligomeric HA-Ycf4 module. In addition to the PSI core subunits, the oligomeric HA-Ycf4 module was associated with LHCI proteins, which were not pulse-labeled (Fig. 4). Subsequent integration of the pre-existing LHCI complexes (LHCA1, LHCA3–8) into the PSI core subcomplex was also assisted by the oligomeric HA-Ycf4 module (Fig. 5). It is possible that a small amount of free LHCI complexes is stably present in the thylakoid membranes because oligomeric LHCI complexes accumulate in the thylakoid membranes of a PSI-deficient mutant[32].

Collectively, we present a PSI complex assembly model (Fig. 6). Ycf3, Y3IP1, and Ycf4 constitute the main part of the assembly apparatus which mediates almost whole assembly steps of PSI and LHCI subunits into a mature PSI–LHCI supercomplex. Since Ycf3 turned over rapidly and Y3IP1 was stable, a newly synthesized Ycf3 and a pre-existing Y3IP1 form the Ycf3–Y3IP1 module (stage I). It is accepted that PsaB is first and PsaA is subsequently synthesized[33], and they are co-translationally inserted into the thylakoid membranes by the assist of some factors, such as Alb3 (Albino3)[34]. The Ycf3–Y3IP1 module assists an initial assembly of PsaA and PsaB into a PSI RC subcomplex (stage II). The TPR motifs in Ycf3, which is supposed to facilitate protein-protein interaction, may play a pivotal role in the initial assembly process. This finding is in agreement with the reports that the deletion of *ycf3* gene completely blocked the accumulation of PSI complex in *C. reinhardtii*[9] and tobacco[20]. PSI auxiliary proteins, PPD1[12] and PSA2[13], which are located on the lumenal face of the thylakoid

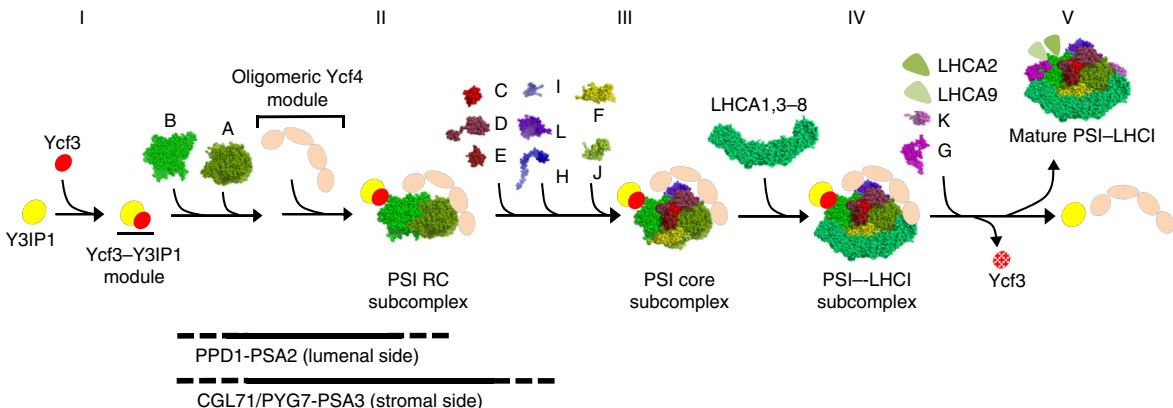

**Fig. 6** Proposed model of PSI complex assembly. Newly synthesized Ycf3 associates with stable Y3IP1 to form a Ycf3–Y3IP1 module in the thylakoid membranes (stage I). This module mediates the assembly of newly synthesized PsaB and PsaA into a reaction center (RC) heterodimer (stage II). PPD1and PSA2 may assist PSI RC assembly from the lumenal side, whereas CGL71/PYG7 on the stromal side together with PSA3 may assists oxygen sensitive assembly steps. Oligomeric Ycf4 module stabilizes newly synthesized PSI RC subcomplex (stage II) and subsequently facilitates the integration of other PSI core subunits, i.e., PsaC, PSAD-F, H, I, J, L to the PSI RC subcomplex to form a PSI core subcomplex (stage III). Seven pre-existing LHCAs (except for LHCA2/9) are integrated to the core subcomplex to form a PSI-LHCI subcomplex on the Ycf4 oligomeric module (stage IV). Upon the integration of PSAG/K and LHCA2/9 to the PSI–LHCI subcomplex, the resulting mature PSI–LHCI supercomplex is released from the assembly apparatus (stage V). Y3IP1 and Ycf4 are reused for the subsequent PSI complex assembly while Ycf3 is replaced with newly synthesized one

membranes, may be involved in the initial assembly steps of PSI RC, although they were not detected in the Ycf3–Y3IP1 preparation. PPD1 interacts with some luminal loops of PsaA and PsaB polypeptides based on the results obtained by yeast two-hybrid assay[12]. PSA2, which is a DnaJ-type zinc finger protein and has protein-disulfide isomerase activity, is also essential in PSI RC accumulation[13]. These two factors may be required for the proper assembly of two RC subunits.

Upon the assembly of the PSI RC subcomplex, the oligomeric Ycf4 module binds to Ycf3–Y3IP1–PSI RC subcomplex (stage II), which may facilitate the integration of other peripheral PSI proteins into the PSI RC subcomplex to form as PSI core subcomplex probably by stabilizing assembly intermediates (stage III). However, it cannot be excluded that the Ycf3–Y3IP1 module is also involved in the integration of some peripheral PSI proteins[21], although sufficient evidence for this process was not obtained in the present study. A similar PSI core subcomplex has already been identified in WT cells at logarithmic growth phase[8]. This assembly process can proceed, although inefficiently, without Ycf4 in a cyanobacterium[19] and tobacco[23] in which the PSI assembly intermediates might be more stable. In addition, PYG7[10]/CGL71[17] in association with PSA3[14] may cooperate for PSI assembly on the stromal face. CGL71 is not essential in the PSI complex assembly in anoxic conditions but is exclusively required for the PSI complex assembly under oxic conditions[17]. However, these factors were not detected in the Ycf3–Y3IP1 and the oligomeric Ycf4 preparations, suggesting a weak physical interaction between them. In the next assembly step, LHCI complexes (LHCA1, 3–8) are integrated into the PSI core subcomplex to form PSI–LHCI subcomplex (stage IV). This step is facilitated by the integration of PSAF into the PSI core subcomplex[35]. Since the LHCI proteins in the oligomeric Ycf4 module were not pulse-labeled, they pre-exist as an oligomeric form as detected in the PSI-deficient mutant[32]. The last assembly steps are the integration of PSAG and PSAK, which are proposed to stabilize the LHCI oligomer association with PSI core[8], as well as LHCA2 and LHCA9[32]. The mature PSI–LHCI supercomplex should be dissociated from the assembly apparatus upon the integration of these proteins (stage V). Ycf4 and Y3IP1 are stable and reutilized in the assembly process, whereas Ycf3 may not be reused because it turns over rapidly for an unknown reason.

Factors responsible for the integration of various cofactors to the PSI complex were not detected in the Ycf3–Y3IP1 and the oligomeric Ycf4 modules obtained under the present purification conditions. Further investigations are required to elucidate the cofactor integration process.

## Methods

**Strains and growth conditions**. Wild-Type 137c (WT) and mutant strains of *C. reinhardtii* were grown to $2–4 \times 10^6$ cells mL$^{-1}$ in tris-acetate-phosphate (TAP) or High-Salt-Minimum (HSM) media under continuous light conditions (2 or 50 µmol photons m$^{-2}$ s$^{-1}$). Chloroplast transformants were selected on HSM solid medium under photoautotrophic growth conditions (50 µmol photons m$^{-2}$ s$^{-1}$) or on TAP solid medium containing 150 µg mL$^{-1}$ spectinomycin. ΔY3IP1 mutant and its WT host strain were obtained from Chlamydomonas Library Project (CLiP)[30]. For growth curves, pre-cultures were grown to mid-log phase ($2–3 \times 10^6$ cells mL$^{-1}$) in TAP media and were inoculated to 50 mL of HSM media at the density of $1–4 \times 10^5$ cells mL$^{-1}$ (0.25$^{-1}$ µg Chl mL$^{-1}$). All the cultures were continuously aerated in the light at 50 µmol photons m$^{-2}$ s$^{-1}$. Growth curves from three biological replicates were analyzed, and typical results are shown in Supplementary Figs. 1c and 3c.

**Chloroplast transformation**. To drive the expression of *ycf3* and *ycf4* in the chloroplast, we used the chloroplast expression vector containing P/5′UTR of *psaA-exon1* and the 3′-UTR of *rbcL* as already described[25]. A *Stu*I restriction site (AGGCCT) was inserted at six nucleotides upstream of the initiation codon in the P/5′-UTR. The resulting vector was digested with *Stu*I and *Sph*I. *Sph*I-digested PCR product containing the eight nucleotides upstream of the *psaA* initiation codon and the coding sequence of *ycf3* or *ycf4* was inserted to generate *ycf3* or *ycf4* expression vector. Two sets of primers, P1/P2, and P3/P4 (Supplementary Table 2) were used to amplify *ycf3* and *ycf4*, respectively, using WT total DNA as a template. The nucleotide sequence coding for HA epitope (human influenza virus hemagglutinin) tag (YPYDVPDYA) was optimized for chloroplast codon usage. To fuse the nucleotide sequence for HA epitope tag before the stop codon of the *ycf3* coding sequence, a synthetic fragment was generated by annealing a pair of oligonucleotides, P5/P6 (Supplementary Table 2), that had been phosphorylated at their 5′ ends and then was ligated between *Aat*II and *Afl*II restriction sites in the *ycf3* expression vector. To fuse the nucleotide sequence for HA epitope tag after the initiation codon of the *ycf4* coding sequence, primers containing HA nucleotide sequence (P7/P8), were designed for inverse PCR, and the amplified inverse PCR product from the *ycf4* expression vector described above as a template was self-ligated. The resulting chimeric gene cassettes were separately inserted at the *Bam*HI site downstream of the *psbA* gene to construct transformation vectors as described[29].

The transformation vectors coated on tungsten particles were delivered into chloroplast of *Chlamydomonas* Fud7 strain which has a deletion in the *psbA* gene as described[29]. Putative transformants were selected under photoautotrophic growth conditions. The control strain was generated by complementation with the transformation vector without the chimeric cassette. Transformants were subjected

to three rounds of single colony purification. Endogenous *ycf3* or *ycf4* gene was disrupted with the *aadA* cassette, in the ycf3-HA and HA-ycf4 transformants, respectively, to generate ycf3-HA/Δycf3 and HA-ycf4/Δycf4 transformants as described[9].

**Nuclear transformation**. We obtained ΔY3IP1 mutant (LMJ.RY0402.195677) from Chlamydomonas Library Project (https://www.chlamylibrary.org), in which paromomycin resistance cassettes (CIB1) were inserted at exon 7 of *Y3IP1*(Cre06. g280650) and *CAH2* (Cre04.g223050)[30]. *CAH2* is functional under low $CO_2$ conditions and should not affect photosystems. The ΔY3IP1 mutant was complemented with the expression vector, pSL18[36], under the control of the promoter and terminator of *PSAD*. The paromomycin resistance gene in this vector was replaced by the hygromycin resistance gene from pHyg3 (Chlamydomonas Resource Center), which was designated as pSL18-Hyg3. The cDNA of *Y3IP1* was amplified from cDNA library of *Chlamydomonas*[26] by using primers, P13/P14 (Supplementary Table 2), and inserted into the pSL18-Hyg3 digested with *Nde*I and *Xba*I. The resulting transformation vector was designated as pSL18-Hyg3-Y3IP1. A DNA fragment consisting of the nucleotide sequence of HA epitope (optimized to *Chlamydomonas* nuclear codon usage), together with the sequence of *Bst*Z17I restriction site, was generated by annealing a set of nucleotides, P15/16 (Supplementary Table 2), and was cloned into pUC18 digested with *Eco*RI and *Xba*I (pUC18-HA). 329 nucleotides of the *Y3IP1* coding sequence with 15 nucleotides of pUC18-HA before *Bst*Z17I restriction site at 5′ end and 15 nucleotides of 5′ end of the sequence for HA tag was amplified by PCR with a pair of primers, P17/18, and cDNA library as a template. The resulting PCR product was inserted into the pUC18-HA digested with *Bst*Z17I using an In-Fusion HD Cloning Kit (Clontech). The resulting construct was digested with *Mlu*I and *Xba*I, and the fragment containing 3′ end of the *Y3IP1-HA* coding sequence was inserted into the pSL18-Hyg3-Y3IP1 digested with *Mlu*I and *Xba*I. The resulting vector, pSL18-Hyg3-Y3IP1-HA, was used to express Y3IP1 fused with HA tag at its C-terminus.

Nuclear transformation was carried out by electroporation with an electroporator (Super Electroporator, NEPA21 typeII, Nepagene, Japan)[37]. ΔY3IP1 cells were harvested at $600 \times g$ and resuspended in TAP-sucrose to the density of $1 \times 10^8$ cells mL$^{-1}$. The transforming vector linearized by digestion with *Not*I (400 ng) was mixed with 38 μL of the cell suspension for electroporation. Putative transformants were selected on TAP agar plates in the presence of hygromycin (10 μg mL$^{-1}$) and then on HSM agar plates in the light (50 μmol photons m$^{-2}$ s$^{-1}$).

**Preparation of thylakoids and chlorophyll–protein complexes**. WT and mutant strains grown in TAP medium to the density of $2–3 \times 10^6$ cells mL$^{-1}$ were centrifuged at $2860 \times g$ for 5 min, and thylakoids were prepared as described by Ozawa et al.[8]. Thylakoids (0.8 mg Chl mL$^{-1}$) solubilized with 1.0% (w/v) *n*-dodecyl-β-D-maltoside (β-DM) or *n*-dodecyl-α-D-maltoside (α-DM) were loaded on to sucrose density gradients (0.1–1.3 M sucrose, 5 mM Tricine-NaOH pH 8.0, 0.05% β-DM/α-DM) and centrifuged at $274,000 \times g$ (SW 41 Ti; Beckman coulter, USA) at 4 °C for 16 h to separate chlorophyll–protein complexes. The gradients were fractionated and analyzed by immunoblotting.

**Affinity purification of HA-tagged proteins**. Thylakoid membranes (50–200 μg Chl mL$^{-1}$) were solubilized with 1% α-DM (for Ycf3-HA and Y3IP1-HA) or 1% β-DM (for HA-Ycf4) for 10 min and centrifuged at $21,500 \times g$ for 10 min. The resulting extracts (500 μL) were incubated with 20 μL of anti-HA beads in a spin column (HA-tagged protein purification kit, MBL) for 1 h by end-over-end mixing as indicated in the manual with some minor modifications. Non-adsorbed extracts were removed by centrifugation. After the beads were washed with the washing buffer for 3–5 times, proteins were eluted with 40 μL of the elution buffer containing HA peptide with 0.05% (β-DM/α-DM). The purified HA-Ycf4 or Ycf3-HA preparation was separated by SDG ultracentrifugation (0.1–1.3 M sucrose, 5 mM Tricine-NaOH pH 8.0, 0.05% β-DM/α-DM) with an SW50.1 rotor (Beckman) at $202,000 \times g$ for 16 h. The gradients were fractionated and analyzed by SDS-PAGE or immunoblotting. The fractions were concentrated with Amicon ultra 100 K filters (Millipore).

**Pulse-chase labeling of total cellular proteins**. Total cellular proteins were labeled with $^{35}$S as described previously[8]. Pre-cultures grown in TAP media with less sulfur to the cell density of $2–3 \times 10^6$ cells mL$^{-1}$ were harvested, resuspended to 25 μg Chl mL$^{-1}$ in TAP media containing no sulfur, and starved for 2 h. Total cellular proteins were labeled with 5 μCi mL$^{-1}$ [$^{35}$S]Na$_2$SO$_4$ (American Radiolabeled Chemicals) in the light at 50 μmol m$^{-2}$ s$^{-1}$ for 10 min. The labeled proteins were chased in the presence of 10 mM Na$_2$SO$_4$ for 6 h. Immediately after pulse labeling and chase, cells were broken by vigorous vortexing with glass beads, and the thylakoids were purified in a small scale by discontinues sucrose density gradient ultracentrifugation as described by Ozawa et al.[8]. After electrophoresis, gels were dried and exposed to imaging plate for several weeks and the labeled polypeptides were detected with a fluorescent image analyzer (FLA7000, Fujifilm).

**SDS-PAGE and immunoblotting**. Polypeptides were separated by SDS-PAGE according to Laemmli[38] in Figs. 1b, 2, 3c, and 4b, and Supplementary Figs. 4 and 6 and according to high-molarity Tris buffer system[39] in the other figures. Separated polypeptides were stained with Flamingo fluorescent dye (Bio-Rad) and detected with an FLA-7000 fluoro-image analyzer (Fujifilm) or transferred to nitrocellulose membranes to probe with different antibodies such as PsaA (1:10,000), PsaC[9] (1:1000), PSAD[24] (1:2000), PSAE[24] (1:1000), PSAF[24] (1:5000), PSAG[8] (1:1000), PSAH[24] (1:1000), PSAI[24] (1:1000), PSAK[8] (1:1000), PSAL[24] (1:5000), Ycf3[9] (1:1000), Ycf4[9] (1:1000), Y3IP1 (1:1000), COP2[24] (1:1000), LHCA1-9[40] (1:1000) and HA (MBL, CODE No. M180-3, 1:1000). Antibody against Y3IP1 was generated using synthesized oligopeptides, CARERRAQDIEDERGVS and CDNLRE-GEKPQNLTDML, of which cysteine residue was conjugated to the carrier protein, keyhole limpet hemocyanin (Eurogentec). The immunodetection signals were visualized by enhanced chemiluminescence (ECL) with a luminescent image analyzer (LAS-4000 mini, Fuji film). The gel/blot images which are trimmed in this manuscript (Figs. 1b, 4b–d, and 4f) have been uncropped in Supplementary Fig. 7.

**Chl a fluorescence measurement**. Chl a fluorescence kinetics were measured in Chlamydomonas cells by using chlorophyll fluorescence analyzer (Dual PAM 100, WALZ)[41]. Cells grown in liquid TAP medium to a midlog growth phase in the light of 5 μmol photon m$^{-2}$ s$^{-1}$ for mutant strains (Δycf3, Δycf4, and ΔY3IP1), or 50 μmol photon m$^{-2}$ s$^{-1}$ for other strains (control, ycf3-HA, ycf3-HA /Δycf3, HA-ycf4, HA-ycf4/Δycf4, c-Y3IP1, and c-Y3IP1-HA) were dark adapted for 30 min. During the measurements, cells were applied with measuring light (ML, 5 μmol photons m$^{-2}$ s$^{-1}$) for ~ 1 s and followed by the addition of actinic light (AL, 127 μmol photons m$^{-2}$ s$^{-1}$) for 3−4 s to monitor PSI activity.

**Pigment analysis by HPLC and absorption spectroscopy**. Pigments were extracted from the lyophilized HA-Ycf4 and PSI–LHCI preparations with *N,N*-dimethyl-formamide (DMF) and were applied onto an HPLC (ACQUITY, UPLC/PDA system; Waters)[42]. Absorption spectra were recorded with a spectrophotometer (SHIMADZU, UV2450S).

**In-gel digestion and LC-MS/MS**. Polypeptides separated by SDS-PAGE were stained with silver, subjected to in-gel digestion by incubating excised gel slices with trypsin (Sequence grade, Promega) at 37 °C for 16 h, and the resulting polypeptides were applied onto an LTQ ion trap mass spectrometer (Thermofinnigan) as described[24].

**Data availability**. The authors declare that all the data supporting the finding of this study are available within the paper and Supplementary Information. All relevant mutants and antibodies are available from the corresponding author upon request.

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

## Acknowledgements

We thank Prof. M. Goldschmidt-Clermont for the kind gift of cDNA library and pSL18 vector. We thank Prof. Kevin Redding for kindly providing the antibodies against PsaA. This work was supported by the Japan Science and Technology Agency (JST), CREST, and JSPS KAKENHI grant numbers 16H06554.

## Author contributions

S.N. and Y.T. designed the study, S.N. performed molecular biology and biochemical experiments, H.K. assisted mutant generation, and S.I.O. assisted biochemical experiments. S.N. and Y.T. analyzed the data and wrote the manuscript, and all authors approved it.

## Additional information

**Competing interests:** The authors declare no competing interests.

