## [Peer Review File · Nature Communications]

Reviewers' comments:

Reviewer #1 (Remarks to the Author):

The biogenesis of PSI complex in plants and algae involves a coordinated synthesis of nuclear and chloroplast encoded polypeptides along with the cofactors. Previous work identified several auxiliary factors including Ycf3, Ycf4, PYG7, Y3IP1, PPD1, PSA2 and PSA3. In this work the authors used *Chlamydomonas reinhardtii* to clearly show that the extrinsic Ycf3 and the Ycf4 not only required for the assembly of PSI but also present in the assembly apparatus in the thylakoid membrane. The assembly apparatus operates in two stages that were physically isolated. First part that contains Ycf3 and Y3IP1 is required for the assembly of PSI reaction center subunits, PsaA and PsaB. The second part contains an oligomeric Ycf4 that facilitates the integration of other peripheral PSI subunits and LHCI to the PSI reaction center subcomplex.

The work used with high proficiency the best available biochemical techniques. The results are novel, clear and convincing. The conclusions are interesting and well founded. I would highly recommend accepting the manuscript for publication.

Reviewer #2 (Remarks to the Author):

This manuscript reports on the biochemical characterization of chloroplast protein complexes involved in the assembly process of Photosystem I in the green alga *Chlamydomonas reinhardtii*. As compared to Photosystem II, the biogenesis of PSI is less well understood due to its rapid assembly and, therefore, more detailed information on this essential process is clearly required to understand thylakoid membrane biogenesis as a whole.

Starting from the previously identified and characterized PSI assembly factors Ycf 3, its interacting factor Y3IP-1, and Ycf4, the authors generated HA-tagged algal lines for each of these. Next, they thoroughly monitored the composition of respective affinity purified complexes via immune-detection or mass spectrometry. Following this approach they confirmed the tight interaction between Ycf3 and Y3IP-1 which has previously been described for tobacco chloroplasts. Moreover, a weaker interaction of Ycf3/Y3IP-1 and Ycf4 was observed for the first time. Based on the subunit composition of the purified complexes the authors then provide a model of PSI biogenesis predicting the initial steps to be mediated by an Ycf3/Y3IP-1 module whereas later steps are organized via an oligomeric Ycf4 module. Overall, this work appears solid, thorough and comprehensive. However, I have some concerns with regard to the interpretation of data sets considering previously published work. Both Ycf3 and Ycf4 interaction partners in *Chlamydomonas* chloroplasts have already been analyzed but the results and thus also the conclusions differ substantially from this work (Naver et al., 2001; Ozawa et al., 2009). For instance, immunoprecipitation of Ycf3 complexes by using an Ycf3 antibody revealed many additional PSI core subunits in Ycf3 complexes besides PsaA and PsaB including PsaC, PsaD, PsaL. Furthermore, Naver et al., convincingly demonstrated a direct interaction between Ycf3 and PsaD which was not detected in Ycf3 complexes here. This would argue for a function of Ycf3 during later steps of PSI assembly in contrast to what the authors propose. This would also be in line with the author's finding that PsaF co-purifies with Ycf3, however, this is not reflected in their model presented in Fig. 3.

Even more puzzling is the finding that the corresponding author's group has previously published that fewer proteins from *Chlamydomonas* co-purify with Ycf4 based on either a TAP-tag approach or immunoprecipitations by using an Ycf4 antibody (Ozawa et al., 2009). In particular, an interaction with the Lhca proteins was excluded before, in contrast to what is proposed in this manuscript. Taken together, I feel that the authors present solid work but it appears that the experimental set-up for complex isolation strongly affects the outcome with regard to potential interaction partners. Thus, I hesitate to follow the authors on their interpretation of the sequence of events for PSI assembly and

the transient functional roles of Ycf3/Y3IP1 and Ycf4, respectively. In particular, a role of Ycf3/Y3IP1 during later steps must be considered questioning a modular nature of the PSI assembly.

Minor points:

- In previous work, the chlamyopsin COP2 has been identified to form part of an Ycf4-TAP-tag-purified complex. Does this hold also for this preparation?

Reviewer #3 (Remarks to the Author):

Nellaepalli et al are working on the identification of the core polypeptides associated with the assembly of PSI. The paper presents evidence that Ycf3, Y3IP1 and Ycf4 form a core PSI assembly complex. They HA tagged Ycf3, Y3IP1 Ycf4, expressed the proteins (in both WT and the mutant lacking the corresponding introduced polypeptide) and affinity purified the tagged protein to identify interacting proteins. They also examine a mutant in Y3IP1 and showd that similar to the ycf3 mutant, there is little assembly of PSI. The work is generally difficult work and while some of the results are clear, other results are less clear or are presented in a way that is less clear. Also, a lot of the key data, especially some of the sucrose gradient work, is in the extended data set (and should probably be in the main text). I feel that there are a number of issues that should be addressed before this work is published.

1. It would be worth doing functional assays for PSI activity (in addition to photoautotrophic growth) in the various strains to examine PSI activity more directly; although this may not be absolutely necessary it would improve the manuscript and its conclusions.

2. The authors show that the HA tagged versions of the genes rescue the mutant photoautotrophic growth phenotype, but it is best to do a growth curve and not just spot tests, which are difficult to evaluate quantitatively. While a growth curve was done for the HA-Ycf4 and HA-Ycf4/ Δ ycf4 strains, it was not done for the Ycf3-HA strains. The authors should also be consistent in their notation. Is it Ycf3-HA as in Figure 1 or ycf3-HA as in extended data Figure 1 (there are other places in which the authors are not consistent in using upper and lower case).

3. Immunoblots did detect Ycf4 in the Ycf3-HA pull down, but it is not observed in the polypeptide profile of Fig. 1A (only detected using an antibody, and it is not clear how much of it is present relative to Ycf3-HA). Is the ratio very substoichiometric... what is the ratio? The result also doesn't necessarily mean that Ycf4 is directly binding to Ycp3 as suggested. It might be binding to PsaA or PsaB... or to PsaF, and is pulled down along with the assembly intermediate.

4. The interpretation of the sucrose gradient centrifugation in the extended data Figure 2 is not so clear to me. Only a very small amount of the Ycf3-HA seems to co-migrate on the gradient with Ycf4. Most of the Ycf3-HA migrates with Y3IP1 at the very top of the gradient (where there is essentially no Ycf4). I wouldn't feel confident from this data that there is a PSI RC subcomplex (Extended data Fig. 2) in which Ycf3-HA is bound to Ycf4 in a stable complex since such a small amount of Ycf3-HA is found in that region of the gradient... and it is also spread throughout the gradient (from the 700 kDa band to its peak at the top of the gradient in the LHCII-M region). So is it really clear that Ycf3 and Ycf4 interact?

5. The authors use the pull downs of Y3IP1-HA (Figure 1d) to show that it co-purifies with both Ycp3 and a small amount of Ycf4, indicating that Y3IP13 and Ycf3 mainly mediate PSI RC assembly... but from this data one can't really tell if Ycf4 is present at amounts that are more or less than Ycf3; is there a more quantitative evaluation of this?

6. In the Ycf4-HA pull downs the authors identified a lot of PSI proteins including the untagged Ycf4 suggesting that Ycf4 works as an oligomer. But does this really have to be the case or can there just be more than one Ycf4 in the assembly complex (with no direct interaction between the the Ycp4 polypeptides). It does however appear that there are a number of PSI polypeptides that co-purify with

Ycf4, and that do not really co-purify with Ycf3, suggesting different roles in the assembly process, as suggested in the model given by the authors, even if they are sometimes present in the same complex.

7. The interpretation of the pulse-chase experiment with Ycf4-HA doesn't look so clear. A lot of the label in PsaA and PsaB seemed to be chased, but not much of the label in PsaC,L,D,F is chased in the 6 h chase time. What do the authors think the reason for this is? Furthermore, in the gradient shown in extended data Figure 5, the HA-Ycf4 is at a low when the PsaA is at its peak... it looks like there are two peaks; the high molecular weight peak and the one below (lower molecular mass) the PSI complex. What is this second peak? How does that correspond to the Y1, Y2 and Y3 peaks in extended data Figure 5? Why not try to keep all of the labeling and nomenclature consistent from the beginning?

8. I am not sure the data says that Ycf3, Y3IP1 and Ycf4 constitute the main part of the assembly apparatus. Certainly they are critical for assembly (they are required), but is there other evidence suggesting that other proteins, that may be in low abundance and less strongly associated with the assembly intermediates, are not required. In fact, as the authors indicate, there do appear to be a number of other assembly factors that are not included in the model that might be part of the core apparatus.

Generally, I feel that the model makes sense given the data but there are a number of issues that should be addressed prior to publication. I feel that better data can be generated, especially a better gel of Figure 2d (which is critical to the argument), there can be a clearer description and presentation of the data as well as a more thoughtful discussion of some of the results; improving these aspects of the manuscript would make the work and its implications more palatable to the reader, and make the model, with its potential caveats, more compelling.

Reviewer #1 (Remarks to the Author):

The biogenesis of PSI complex in plants and algae involves a coordinated synthesis of nuclear and chloroplast encoded polypeptides along with the cofactors. Previous work identified several auxiliary factors including Ycf3, Ycf4, PYG7, Y3IP1, PPD1, PSA2 and PSA3. In this work the authors used *Chlamydomonas reinhardtii* to clearly show that the extrinsic Ycf3 and the Ycf4 not only required for the assembly of PSI but also present in the assembly apparatus in the thylakoid membrane. The assembly apparatus operates in two stages that were physically isolated. First part that contains Ycf3 and Y3IP1 is required for the assembly of PSI reaction center subunits, PsaA and PsaB. The second part contains an oligomeric Ycf4 that facilitates the integration of other peripheral PSI subunits and LHCI to the PSI reaction center subcomplex. The work used with high proficiency the best available biochemical techniques. The results are novel, clear and convincing. The conclusions are interesting and well founded. I would highly recommend accepting the manuscript for publication.

Ans. Thank you very much for the positive comments.

Reviewer #2 (Remarks to the Author):

This manuscript reports on the biochemical characterization of chloroplast protein complexes involved in the assembly process of Photosystem I in the green alga *Chlamydomonas reinhardtii*. As compared to Photosystem II, the biogenesis of PSI is less well understood due to its rapid assembly and, therefore, more detailed information on this essential process is clearly required to understand thylakoid membrane biogenesis as a whole. Starting from the previously identified and characterized PSI assembly factors Ycf 3, its interacting factor Y3IP-1, and Ycf4, the authors generated HA-tagged algal lines for each of these. Next, they thoroughly monitored the composition of respective affinity purified complexes via immune-detection or mass spectrometry. Following this approach they confirmed the tight interaction between Ycf3 and Y3IP-1 which has previously been described for tobacco chloroplasts. Moreover, a weaker interaction of Ycf3/Y3IP-1 and Ycf4 was observed for the first time. Based on the subunit composition of the purified complexes the authors then provide a model of PSI biogenesis predicting the initial steps to be mediated by an Ycf3/Y3IP-1 module whereas later steps are organized via an oligomeric Ycf4 module. Overall, this work appears solid, thorough and comprehensive.

However, I have some concerns with regard to the interpretation of data sets considering previously published work. Both Ycf3 and Ycf4 interaction partners in *Chlamydomonas* chloroplasts have already been analyzed but the results and thus also the conclusions differ substantially from this work (Naver et al., 2001; Ozawa et al., 2009). For instance, immunoprecipitation of Ycf3 complexes by using an Ycf3 antibody revealed many additional PSI core subunits in Ycf3 complexes besides PsaA and PsaB including PsaC, PsaD, PsaL.

Ans: Naver et al. (2001) immunoprecipitated PSI core complexes from the thylakoid extracts solubilized with β -DM by using a Ycf3 antibody. Their results indicate that Ycf3 has an interaction with isolated PSI core complex consisting PsaA, PsaB, PsaC, PSAD, and PSAL. However, it remains elusive whether the immunoprecipitated PSI complexes are newly synthesized or not. In addition, the previous results (Albus et al. 2010) and the present results revealed that both Ycf3 and Y3IP1, which form the complex, are required for the PSI

complex assembly. Thus, we think that the interaction between Ycf3 and PSI complexes in the absence of Y3IP1 shown by Naver et al. is not sufficient to conclude that Ycf3 is involved in the PSI core complex assembly. As described in the manuscript, we observed the interaction of Ycf3-Y3IP1 module with newly synthesized PSI RC subunits but not with other PSI subunits, strongly suggesting that at least the Ycf3-Y3IP1 module is involved in the initial PSI RC subcomplex assembly. However, we could not obtain evidence supporting that the Ycf3 is required for the subsequent assembly steps. Nonetheless, we cannot exclude the possibility that the peripheral PSI proteins might be lost from the Y3IP1 preparation during the affinity purification although we employed rather a mild purification procedure in the present study. Thus, in the PSI complex assembly model as shown in Fig. 6, the Ycf3-Y3IP1 module is located with the oligomeric Ycf4 and the assembly intermediates (Fig. 6 stages III and IV). (This information has been included in the main text; lines 54-57, 101-112, and 299-320).

Furthermore, Naver et al., convincingly demonstrated a direct interaction between Ycf3 and PsaD which was not detected in Ycf3 complexes here. This would argue for a function of Ycf3 during later steps of PSI assembly in contrast to what the authors propose.

Ans: Naver et al. (2001) showed an interaction of recombinant His-tagged Ycf3 with denatured PsaA blotted onto a nitrocellulose membrane, and with PSAD extracted from the thylakoid membranes. These results indicate the interactions under *in vitro* conditions, whereas we showed an interaction of Ycf3 with newly synthesized PsaA/B under more natural conditions. As described above, although we do not exclude the possibility that the Ycf3-Y3IP1 module lost PSAD during the purification, it remains elusive whether the interaction between the His-tagged Ycf3 and the extracted PSAD in the absence of Y3IP1 is sufficient evidence for the involvement of the Ycf3-Y3IP1 module in the assembly of the peripheral PSI proteins. Accordingly, we have revised the manuscript and also added Ycf3-Y3IP1 module in Fig. 6 as described above. (This information has been included in the main text; lines 54-57, and 299-320).

This would also be in line with the author's finding that PsaF co-purifies with Ycf3, however, this is not reflected in their model presented in Fig. 3.

Ans: The PSAF in the Ycf3-HA preparation was neither integrated to the main PSI RC subcomplex (700 kDa) nor radiolabeled (Fig 2 and Fig. 1e). It is noted that PSAF accumulated at a low level in Δ PsaA/B mutant, was associated with Ycf4 (Ozawa et al. 2009), suggesting that the PSAF in the Ycf3-HA affinity purification could be more preferentially bound to the Ycf4 copurified with the Ycf3-HA. Thus, we concluded that the Ycf3-Y3IP1 module is not involved in the integration of PSAF into a PSI RC subcomplex, whereas, PSAF in HA-Ycf4 module is integrated to PSI core and PSI-LHCI subcomplexes, indicating its integration during later steps of PSI assembly as described earlier in tobacco by Wittenberg et al. 2017 (This information has been included in the main text; lines 123, 139-143, 313-317, and 371-372).

Even more puzzling is the finding that the corresponding author's group has previously published that fewer proteins from *Chlamydomonas* co-purify with Ycf4 based on either a TAP-tag approach or immunoprecipitations by using an Ycf4 antibody (Ozawa et al., 2009). In particular, an interaction with the Lhca proteins was excluded before, in contrast to what is proposed in this manuscript.

Ans: We have revised the manuscript to explain why the HA-Ycf4 preparation associated more PSI proteins and also Lhca proteins. The main reason for the difference is that the affinity purification of the HA-Ycf4 completed within two hours although the two-step affinity purification of Ycf4-TAP took one day. The shorter purification method is milder to the preparation so that some weakly-bound PSI proteins and Lhca proteins were efficiently co-purified with HA-Ycf4. In the previous report, the immunoprecipitations of labeled PSI proteins using the Ycf4 antibody revealed us that only PSAD/PSAF were transiently labeled, but in the present study, we could show that PsaA/B, PsaC, PSAD/PSAF, and PSAL with higher resolutions and intensities. (This information has been included in the main text; lines 163-172, 197-222, and 251-254).

Taken together, I feel that the authors present solid work but it appears that the experimental set-up for complex isolation strongly affects the outcome with regard to potential interaction partners. Thus, I hesitate to follow the authors on their interpretation of the sequence of events for PSI assembly and the transient functional roles of Ycf3/Y3IP1 and Ycf4, respectively. In particular, a role of Ycf3/Y3IP1 during later steps must be considered questioning a modular nature of the PSI assembly.

Ans: In the present study, we revealed that the affinity-purified HA-Ycf4 associated more PSI proteins and LHCI proteins than the affinity purified TAP-Ycf4. Since our new results on affinity purification under milder conditions provided us improved information rather than conflicting findings. Apparently, we could obtain the HA-Ycf4 with a higher yield, and as a result, identification of PSI and LHCI polypeptides was much easier and reliable as well as pulse-labeling was much easier and more precise. This was also the case for Ycf3 preparation. We have revised the working model for PSI complex assembly. (This information has been included in the main text; lines 163-172, 197-222, and 251-254).

Minor points:

- In previous work, the chlamyopsin COP2 has been identified to form part of an Ycf4-TAP-tag- purified complex. Does this hold also for this preparation?

Ans: COP2 and PSAF were major polypeptides in the Ycf4-TAP preparation although COP2 was not essential for PSI accumulation as reported in the previous work. A small amount of COP2 was detected not only in the HA-Ycf4 preparation but also in the preparation from WT in the present work, suggesting its nonspecific interaction (Fig 4d, e). Since longer incubation of the thylakoid extracts with affinity beads increases non-specific interactions, it is possible that COP2 could be enriched due to the longer purification (overnight incubation) with the affinity resins during the TAP-tag affinity preparation. (This information has been included in the main text; lines 209-222)

Reviewer #3 (Remarks to the Author):

Nellaepalli et al are working on the identification of the core polypeptides associated with the assembly of PSI. The paper presents evidence that Ycf3, Y3IP1 and Ycf4 form a core PSI assembly complex. They HA tagged Ycf3, Y3IP1 Ycf4, expressed the proteins (in both WT and the mutant lacking the corresponding introduced polypeptide) and affinity purified the tagged protein to identify interacting proteins. They also examine a mutant in Y3IP1 and showd that similar to the ycf3 mutant, there is little assembly of PSI. The work is generally

difficult work and while some of the results are clear, other results are less clear or are presented in a way that is less clear.

Ans: Thank you for the valuable comments and suggestions. We have extensively revised our entire manuscript to describe the results in more detail. We have added new data, i.e., growth curves for *ycf3*-HA strains and fluorescence kinetics to show PSI activity, and we have rearranged some of the main Figures with data from Supplementary Figures and revised our model on PSI assembly.

Also, a lot of the key data, especially some of the sucrose gradient work, is in the extended data set (and should probably be in the main text).

Ans:

- Extended Data Fig 1a, c has been moved to Fig 1a, b.
- Extended Data Fig 2 has been moved to Fig 2.
- Extended Data Fig 3 and Fig. 1d, e has been moved to Fig. 3.
- Extended Data Fig 4a, Extended Data Fig 6 and Fig. 2 a, b has been moved to Fig 4.
- Extended Data Fig 8a and Fig 2 c, d has been moved to Fig 5.

I feel that there are a number of issues that should be addressed before this work is published.

1. It would be worth doing functional assays for PSI activity (in addition to photoautotrophic growth) in the various strains to examine PSI activity more directly; although this may not be absolutely necessary it would improve the manuscript and its conclusions.

Ans: Thank you for your valuable suggestion. We have provided data for PSI activity (Supplementary Fig. 1b, Supplementary Fig. 2 and Supplementary Fig. 3b)

2. The authors show that the HA tagged versions of the genes rescue the mutant photoautotrophic growth phenotype, but it is best to do a growth curve and not just spot tests, which are difficult to evaluate quantitatively. While a growth curve was done for the HA-Ycf4 and HA-Ycf4/ Δ ycf4 strains, it was not done for the Ycf3-HA strains.

Ans: The growth curve for *ycf3*-HA and *ycf3*-HA/ Δ ycf3 has been provided as Supplementary Fig 1c.

The authors should also be consistent in their notation. Is it Ycf3-HA as in Figure 1 or *ycf3*-HA as in extended data Figure 1 (there are other places in which the authors are not consistent in using upper and lower case).

Ans: We used *ycf3*-HA to indicate *ycf3* expression cassette, *ycf3*-HA (*ycf3*-HA mutant) to indicate strain name and Ycf3-HA to indicate Ycf3 protein tagged with HA.

3. Immunoblots did detect Ycf4 in the Ycf3-HA pull down, but it is not observed in the polypeptide profile of Fig. 1A (only detected using an antibody, and it is not clear how much of it is present relative to Ycf3-HA). Is the ratio very substoichiometric... what is the ratio? The result also doesn't necessarily mean that Ycf4 is directly binding to Ycf3 as suggested. It might be binding to PsaA or PsaB... or to PsaF, and is pulled down along with the assembly intermediate.

Ans. Thank you very much for the valuable comments and suggestions. We have revised the manuscript to describe that Ycf4 in the Ycf3-HA preparation was present at a substoichiometric level. However, it is difficult to estimate the ratio because the Ycf4 was not detected by staining. We don't have evidence to show a direct interaction between Ycf3 and Ycf4 and agree the possibility that the interaction could be indirect via PsaA/B. As suggested, we revised the manuscript and the model in Fig. 6 to describe that Ycf4 may not directly bind to Ycf3. (This reflects in the revised PSI assembly model).

4. The interpretation of the sucrose gradient centrifugation in the extended data Figure 2 is not so clear to me. Only a very small amount of the Ycf3-HA seems to co-migrate on the gradient with Ycf4. Most of the Ycf3-HA migrates with Y3IP1 at the very top of the gradient (where there is essentially no Ycf4). I wouldn't feel confident from this data that there is a PSI RC subcomplex (Extended data Fig. 2) in which Ycf3-HA is bound to Ycf4 in a stable complex since such a small amount of Ycf3-HA is found in that region of the gradient... and it is also spread throughout the gradient (from the 700 kDa band to its peak at the top of the gradient in the LHCII-M region). So is it really clear that Ycf3 and Ycf4 interact?

Ans. Thank you for the valuable comments. We agree that the result in Fig. 2 in the revised manuscript (extended data Figure 2 in the previous manuscript) does not give us the evidence for a direct interaction between the Ycf3-Y3IP1 complex and Ycf4 because the Ycf3-HA-Y3IP1 preparation contained the PSI RC subcomplex. Figure 2 shows that the interaction between Ycf3-HA and PsaA/PsaB was weak and was dissociated during SDG ultracentrifugation. It appears that a small amount of Ycf3-HA and Ycf4 bind the PSI reaction center more tightly. Accordingly, we revised the manuscript. (This information has been included in the main text; lines 107-109, 131-143, and Fig 2)

5. The authors use the pull downs of Y3IP1-HA (Figure 1d) to show that it co-purifies with both Ycf3 and a small amount of Ycf4, indicating that Y3IP13 and Ycf3 mainly mediate PSI RC assembly... but from this data one can't really tell if Ycf4 is present at amounts that are more or less than Ycf3; is there a more quantitative evaluation of this?

Ans: As we mentioned above, Ycf4 is present in a substoichiometric level in Ycf3-HA preparation so that it was difficult to detect a stained band by SDS-PAGE. We described this fact in the revised manuscript. (This information has been included in the main text; lines 107-109).

6. In the Ycf4-HA pull downs the authors identified a lot of PSI proteins including the untagged Ycf4 suggesting that Ycf4 works as an oligomer. But does this really have to be the case or can there just be more than one Ycf4 in the assembly complex (with no direct interaction between the the Ycp4 polypeptides).

Ans: We have already reported that Ycf4 is part of a large complex, but it was not so clear how Ycf4 constitutes a large complex (Ozawa et al. Plant Cell, 2009). SDG ultracentrifugation separated Y3 containing almost only Ycf4 at the apparent size >700 kDa below Y2 which is about 700 kDa indicating that Ycf4 form an oligomeric structure of many copies. But it remains unclear how many copies are present in the Ycf4-HA preparation. (This information has been included in the main text; lines 186-194, and 267-275)

It does however appear that there are a number of PSI polypeptides that co-purify with Ycf4, and that do not really co-purify with Ycf3, suggesting different roles in the assembly process,

as suggested in the model given by the authors, even if they are sometimes present in the same complex.

Ans: Thank you very much for the important comments. As described in the present work, the Ycf3-HA preparation contained only a small amount of Ycf4 while HA-Ycf4 preparation contained only a small amount of Ycf3. Thus, we believe that it was justified to conclude the different roles of Ycf3 and Ycf4 from the characterization of the Ycf3 and Ycf4 preparations.

7. The interpretation of the pulse-chase experiment with Ycf4-HA doesn't look so clear. A lot of the label in PsaA and PsaB seemed to be chased, but not much of the label in PsaC, L, D, F is chased in the 6 h chase time. What do the authors think the reason for this is?

Ans: Thank you for the valuable suggestions. We estimated the signal intensities of the label in PsaA/PsaB and the other PSI subunits (PsaC, PSAD/PSAF, and PSAL). It was roughly estimated that the pulse-labeled PsaA/PsaB decayed to approximately 30% after the 6 h chase, whereas peripheral PSI proteins to approximately 50%. We speculated that PsaA/PsaB are bound to the Ycf3-Y3IP1-Ycf4 complex upon their synthesis while other PSI proteins are integrated into the PSI RC subcomplex for a while after the synthesis. If it is assumed that the peripheral PSI proteins pre-exist in a small pool before the assembly, the decay of the labeled proteins should be slower than that of the RC proteins because the pre-existing PSI proteins are a mixture of labeled and non-labeled proteins. (This information has been included in the main text; lines 234-247).

Furthermore, in the gradient shown in extended data Figure 5, the HA-Ycf4 is at a low when the PsaA is at its peak... it looks like there are two peaks; the high molecular weight peak and the one below (lower molecular mass) the PSI complex. What is this second peak?

Ans: PsaA was identified as a major peak at A3 position which corresponds to PSI-LHCI (A3) and was also identified as a minor peak in the bottom heavier fractions which corresponds to PSI-LHCI-LHCII as reported by Takahashi et al. Plant and Cell Physiology 2014 (Supplementary Fig. 4, in the revised manuscript). (This information has been included in the main text; lines 190-192).

How does that correspond to the Y1, Y2 and Y3 peaks in extended data Figure 5? Why not try to keep all of the labeling and nomenclature consistent from the beginning?

Ans: The reason why we used the different nomenclature for the HA-Ycf4 preparations is that Y1, Y2, and Y3 were separated from the affinity-purified HA-Ycf4 preparation and these fractions are different from A2 and A3 from the WT thylakoid extracts (Fig. 5a, in the revised manuscript).

8. I am not sure the data says that Ycf3, Y3IP1 and Ycf4 constitute the main part of the assembly apparatus. Certainly they are critical for assembly (they are required), but is there other evidence suggesting that other proteins, that may be in low abundance and less strongly associated with the assembly intermediates, are not required. In fact, as the authors indicate, there do appear to be a number of other assembly factors that are not included in the model that might be part of the core apparatus.

Ans: Thank you for the important comments. We have included other PSI auxiliary factors, PPD1, PSA2, CGL71/PYG7, and PSA3 in Fig. 6. We have also explained these factors in the

Discussion section. (This information has been included in the main text; lines 349-355, 365-368).

Generally, I feel that the model makes sense given the data but there are a number of issues that should be addressed prior to publication. I feel that better data can be generated, especially a better gel of Figure 2d (which is critical to the argument), there can be a clearer description and presentation of the data as well as a more thoughtful discussion of some of the results; improving these aspects of the manuscript would make the work and its implications more palatable to the reader, and make the model, with its potential caveats, more compelling.

Ans: Thank you for the valuable comments. We intensively tried to present the results in Fig. 2d (Fig 5 c and d in the revised manuscript) and we believe that the signals are sufficient. In the revised manuscript, this data (Fig. 2d) has been moved to Figure 5c, d and rearranged to make the manuscript more understandable. When the purified HA-Ycf4 was separated on SDG, the resulting fractions (Y1-Y3) were very diluted, and thus further characterization of these fractions by immunoblotting is not easy. We have already repeated the experiments to get these data (Fig, 5c, and d). Now the model in Fig, 6 has been revised and improved to make the working hypothesis for the PSI assembly process clearer as described above. Furthermore, we have extensively revised the entire manuscript.

REVIEWERS' COMMENTS:

Reviewer #2 (Remarks to the Author):

In the revised version of the manuscript the authors have appropriately addressed the points I had raised. They now discuss the data with regard to previous work and they have modified their model accordingly. In conclusion, I would recommend to publish this interesting work.

Reviewer #3 (Remarks to the Author):

The work relies heavily on the use of HA tagged PSI assembly factors for affinity purification, sucrose gradient centrifugation and pulse-chase experiments to identify protein-protein interactions to define the sequence of events in the assembly of PSI (through the capture of assembly intermediates and defining the proteins present in these subcomplex intermediates). This is difficult work because of the use of detergents, the time to isolate the intermediates, the lability of the protein-protein interactions etc. However, I think that Nellapaepalli et al. have done the work in a professional thoughtful way and at the end have proposed a very reasonable model. They have also now rewritten some parts of the paper to make their description and interpretation of the results clearer (there are still a few minor wording/English mistakes which I am sure the editor will pick up) and explain why the results sometimes do not exactly align with what was concluded in earlier work (e.g. Naver et al., 2001). While I enjoyed the paper during the first reading, although I did think that some of the explanations were not that clearly written, the results in this improved manuscript were more clearly presented and discussed, with some new supplementary data that also helps. The hypothesis of the assembly involving two modules seems reasonable given the results, although I am sure that it is a little more complicated (e.g. other factors/cofactors must integrate into the assembly process, as does the association of pigments, and the transition from module 1 to 2 in assembly may be more seamless than indicated). In any case, I feel that the authors have addressed most of the questions raised by the reviewers and that the paper is now easier to read/follow and provides keen insights into the assembly of PSI with the use of strong classical biochemical technologies (performed very well).

Reviewer #2 (Remarks to the Author):

In the revised version of the manuscript the authors have appropriately addressed the points I had raised. They now discuss the data with regard to previous work and they have modified their model accordingly. In conclusion, I would recommend to publish this interesting work.

Ans. Thank you for your comments.

Reviewer #3 (Remarks to the Author):

The work relies heavily on the use of HA tagged PSI assembly factors for affinity purification, sucrose gradient centrifugation and pulse-chase experiments to identify protein-protein interactions to define the sequence of events in the assembly of PSI (through the capture of assembly intermediates and defining the proteins present in these subcomplex intermediates). This is difficult work because of the use of detergents, the time to isolate the intermediates, the lability of the protein-protein interactions etc. However, I think that Nellapaepalli et al. have done the work in a professional thoughtful way and at the end have proposed a very reasonable model. They have also now rewritten some parts of the paper to make their description and interpretation of the results clearer (there are still a few minor wording/English mistakes which I am sure the editor will pick up) and explain why the results sometimes do not exactly align with what was concluded in earlier work (e.g. Naver et al., 2001). While I enjoyed the paper during the first reading, although I did think that some of the explanations were not that clearly written, the results in this improved manuscript were more clearly presented and discussed, with some new supplementary data that also helps. The hypothesis of the assembly involving two modules seems reasonable given the results, although I am sure that it is a little more complicated (e.g. other factors/cofactors must integrate into the assembly process, as does the association of pigments, and the transition from module 1 to 2 in assembly may be more seamless than indicated). In any case, I feel that the authors have addressed most of the questions raised by the reviewers and that the paper is now easier to read/follow and provides keen insights into the assembly of PSI with the use of strong classical biochemical technologies (performed very well).

Ans. Thank you for accepting this manuscript.